# Motor Imagery EEG Classification Based on Multi-Domain Feature Rotation and Stacking Ensemble

**DOI:** 10.3390/brainsci15010050

**Published:** 2025-01-07

**Authors:** Xianglong Zhu, Ming Meng, Zewen Yan, Zhizeng Luo

**Affiliations:** School of Automation, Hangzhou Dianzi University, Hangzhou 310018, China; zxl18855428167@163.com (X.Z.); zy12451@163.com (Z.Y.); luo@hdu.edu.cn (Z.L.)

**Keywords:** electroencephalogram, motor imagery, stacking ensemble, multi-domain features, rotation transform

## Abstract

Background: Decoding motor intentions from electroencephalogram (EEG) signals is a critical component of motor imagery-based brain–computer interface (MI–BCIs). In traditional EEG signal classification, effectively utilizing the valuable information contained within the electroencephalogram is crucial. Objectives: To further optimize the use of information from various domains, we propose a novel framework based on multi-domain feature rotation transformation and stacking ensemble for classifying MI tasks. Methods: Initially, we extract the features of Time Domain, Frequency domain, Time-Frequency domain, and Spatial Domain from the EEG signals, and perform feature selection for each domain to identify significant features that possess strong discriminative capacity. Subsequently, local rotation transformations are applied to the significant feature set to generate a rotated feature set, enhancing the representational capacity of the features. Next, the rotated features were fused with the original significant features from each domain to obtain composite features for each domain. Finally, we employ a stacking ensemble approach, where the prediction results of base classifiers corresponding to different domain features and the set of significant features undergo linear discriminant analysis for dimensionality reduction, yielding discriminative feature integration as input for the meta-classifier for classification. Results: The proposed method achieves average classification accuracies of 92.92%, 89.13%, and 86.26% on the BCI Competition III Dataset IVa, BCI Competition IV Dataset I, and BCI Competition IV Dataset 2a, respectively. Conclusions: Experimental results show that the method proposed in this paper outperforms several existing MI classification methods, such as the Common Time-Frequency-Spatial Patterns and the Selective Extract of the Multi-View Time-Frequency Decomposed Spatial, in terms of classification accuracy and robustness.

## 1. Introduction

Brain–Computer Interface (BCI) is a technology that facilitates communication between the nervous system and external devices by establishing a direct connection between them [1]. Common BCI paradigms include Steady-State Visual Evoked Potentials (SSVEPs) [2], P300 [3], and motor imagery (MI) [4]. MI, due to its significant potential for practical applications, has received widespread attention. Electroencephalogram (EEG) studies have shown that when subjects imagine the movement of their left or right hand, the cerebral cortex generates two distinct rhythmic signals. The EEG rhythmic energy significantly decreases in the contralateral sensorimotor area of the cerebral cortex, while the EEG rhythmic energy increases in the ipsilateral sensorimotor area. This phenomenon is known as Event-Related Desynchronization (ERD) and Event-Related Synchronization (ERS) [5,6]. Based on ERD/ERS phenomena, different body movements can be distinguished [7,8]. To enable the effective functioning of BCIs in MI tasks, accurate classification is essential. However, due to the complexity of the data and potential noise interference, this task becomes challenging [9]. Therefore, how to efficiently analyze and utilize the information in EEG signals to improve MI classification performance has turned into a hot subject in current research.

In recent years, to achieve better performance in MI task recognition, researchers have developed various algorithms. For feature extraction, some studies use methods such as Filter Bank Common Spatial Pattern (FBCSP) [10], Power Spectral Density (PSD) [11], time-frequency analysis [12], and various entropy features [13] to classify MI. However, feature extraction that relies solely on a single domain provides only a limited amount of information, leading to suboptimal extraction results. Existing research shows that features obtained from various domains can provide complementary information for classification and display different sensitivities to a range of MI tasks. [14]. Consequently, it is essential to leverage features from multiple domains to enhance classification performance. Ninghan Li et al. [15] achieved MI classification by extracting suitable wavelet features and CSP features from specific frequency bands and then fusing them. Chunyao Xu et al. [16] proposed a two-stage multi-domain feature extraction method based on sparse representation, which selects and reduces the dimensionality of multi-domain features through sparse representation to obtain more effective features for classification. Yue Zhang et al. [17] extracted multi-scale features from various domains and fused them, and then classified the combined features using an ensemble Linear Discriminant Analysis (LDA) classifier. Their findings demonstrated that multi-domain features can synergize and enhance classification performance. Although the aforementioned methods classify EEG signals by extracting multi-domain features, they merely concatenate and fuse the features from different domains. This approach does not fully allow the effective information contained in each domain to complement each other, which ultimately impacts the enhancement of classification accuracy.

In terms of classification, most research has focused on using a single classifier. However, studies indicate that employing ensemble learning with multiple classifiers can enhance classification performance and robustness compared to relying on just one classifier [18]. This approach of using multiple classifier ensembles has been extensively studied in MI tasks. For instance, Muhammad Ammar Ali et al. [19] proposed a classification algorithm based on ensemble pruning, which prunes out the best classifier combination through the difference of convex algorithm (DCA) to improve classification performance. Jing Luo [20] proposed an ensemble support vector learning (ESVL) classifier that combines the probabilities obtained from different SVM classifiers for the classification of MI. Cili Zuo et al. [21] propose a clustering decomposition-based ensemble learning framework (CDECL) that generates a set of heterogeneous classifiers by clustering and decomposing EEG data. The combination of these classifiers is optimized using a random fractal binary multi-objective fruit fly optimization algorithm. This framework demonstrates superior classification performance across multiple public EEG datasets. Hossein Ahmadi et al. [22] proposed the Correlation-Optimized Weighted Stacking Ensemble (COWSE) model, which integrates 16 classifiers through a weighted stacking method. The optimal classifier combination is selected based on error correlation analysis for classifying MI tasks. However, these studies still rely solely on heterogeneous base classifiers, using the same input feature set to predict outcomes, which fails to fully utilize the potential advantages brought by feature diversity. This limitation may lead to an incomplete understanding of the MI process, thereby affecting the improvement of classification performance.

In response to the aforementioned issues, we propose a new method based on multi-domain feature rotation transformation and stacking ensemble model. By integrating the prediction results of base classifiers from various domains and inputting them into a meta-classifier to obtain the final prediction, we effectively combine multi-domain features with the stacking method. This approach not only fully leverages the complementary information provided by different domain-specific features for classification but also enhances the diversity of the stacking ensemble model, thereby improving the accuracy of MI classification. Additionally, we apply local rotation transformations to the features, which enhances their representational ability and discriminability by partitioning the original feature space into random subspaces and reconstructing the feature space using Principal Component Analysis (PCA) [23], thereby further boosting the classification performance of the model. Finally, the effectiveness of the proposed method is validated on three public EEG datasets.

## 2. Materials and Methods

### 2.1. Data Description

Dataset I: This dataset is from BCI Competition III Data Sets Iva, recording EEG data from five subjects (aa, av, al, aw, ay) without feedback. The data includes three types of MI signals (right foot, left, and right hand). Each set of EEG signals was recorded using 118 channels at a sampling rate of 100 Hz. Each subject performed 140 experiments for each type of MI, totaling 280 experiments. The data composition includes a visual cue phase and an MI phase. During the experiment, subjects sat on a chair with their hands naturally placed on the armrests and performed a 3.5 s MI task as required, followed by a 1.75–2.25 s rest period. The specific experimental timeline is shown in Figure 1a.

Dataset II: This dataset is from BCI Competition IV Dataset Set I, recording EEG data from seven subjects without feedback. Since the EEG data for subjects c, d, and e are artificially synthesized, this paper only discusses the data for subjects a, b, f, and g. The data includes three types of MI signals (left and right hand, foot). Each set of EEG signals was recorded using 59 channels at a sampling rate of 100 Hz. Each subject performed 100 experiments for each type of MI, totaling 200 experiments. In each experiment, a fixed image is shown on the computer screen for 2 s, followed by a fixation cross overlaid with the MI cue for a total of 6 s. The specific experimental timeline is shown in Figure 1b.

Dataset III: This dataset is from BCI Competition IV Dataset 2a, recording EEG data from nine subjects (A01, …, A09) without feedback. The dataset includes four types of MI signals (foot, tongue, left, and right hand). Each set of EEG signals was recorded using the International 10–20 system channel distribution at a sampling rate of 250 Hz. Each subject performed six rounds of MI collection, with each round consisting of 12 experiments for each of the four types of MI, resulting in a total of 288 MI experiments per subject as the training dataset. An equal amount of testing data is also available. The data composition includes a visual cue phase and an MI phase. During the experiment, subjects sat on a chair with their hands naturally placed on the armrests and performed a 4 s MI task as required, followed by a 1.5 s rest period. The specific experimental timeline is shown in Figure 1c.

### 2.2. Overview of the Proposed Method

This paper proposes an MI EEG classification method based on multi-domain feature rotation transformation and a stacking ensemble framework, with the overall flowchart illustrated in Figure 2. Firstly, we extract features from EEG signals in four different domains and then perform feature selection within each domain to identify significant features that are more effective for classification. Subsequently, local rotation transformations are applied to the significant features of each domain to obtain a set of rotated features. By fusing the rotated features and the significant features of each domain, a composite feature set is formed. Finally, a stacking ensemble learning method is adopted, integrating the prediction results of base classifiers trained on composite feature sets from different domains and the discriminant features obtained through LDA dimensionality reduction on original significant features as inputs for the meta-classifier. The final classification result is obtained through the meta-classifier.

### 2.3. Multi-Domain Feature Extraction

Feature extraction aims to obtain effective information from the raw EEG signals that will facilitate classification. We extract features from different domains of EEG for subsequent classification tasks.

#### 2.3.1. Time Domain Features

Time Domain (TD) features are statistical measures derived from EEG signals that reflect relevant information along the time dimension. These features effectively reflect the changes in EEG signals over time, providing a strong intuitive understanding of the temporal dynamics of brain activity. This paper calculates Higuchi’s Fractal Dimension $f_{hfd}$, mean $f_{mean}$, variance $f_{\mathrm{var}}$, and root mean square $f_{rms}$ of EEG signals as TD features. Higuchi’s Fractal Dimension [24] is a nonlinear metric that assesses the dynamics of time-domain waveforms. It quantitatively evaluates signal complexity under both normal and abnormal conditions, allowing for the detection of concealed information in neurophysiological time series with a high degree of sensitivity. First, the EEG signal can be represented as a sequence of discrete values: x(1),⋯,x(T). Based on the EEG signal, a new set of self-similar time series is constructed, each series being $x_{m}^{l}$, which is defined as:(1)$$x_{m}^{l}:x(m),x(m+l),x(m+2l),\ldots x(m+Ql)$$
where m is the initial time, l is the time interval, T is the number of data points $l=1,2,\ldots l_{max}$ and $l_{max}$ is a free tuning parameter. Q is the integer part of $\left(\frac{T-l}{l}\right)$. Then the length of each time series $x_{m}^{l}$ is $L_{m}(l)$:(2)$$L_{m}(l)=\frac{1}{l}\left(\sum_{i=1}^{Q}\left|x(m+il)-x(m+(i-1)l)\right|\right)\left(\frac{T-1}{Q}\right)$$

Then, the Higuchi’s Fractal Dimension feature $f_{hfd}$ can be obtained as:(3)$$f_{hfd}=\frac{ln\sum_{m=1}^{l}\frac{L_{m}(l)}{l}}{ln\frac{1}{l}}$$

The final obtained subset of TD features: $F_{td}=\left\{f_{mean},f_{var},f_{rms},f_{hfd}\right\}$

#### 2.3.2. Frequency Domain Features

In Frequency Domain (FD) analysis, the distribution of signal bands in the FD can be obtained from the signal’s spectrogram. Changes in the signal can be derived from changes in the frequency bands. In the FD analysis of EEG signals, it is often required to perform a Discrete Fourier Transform (DFT) on the original time series [25]:(4)$$X(k)=\sum_{t=1}^{T}x(t)e^{-j\frac{2\pi }{T}kt},k=1,2,\ldots T$$
where X(k) represents the k-th frequency component of the EEG output FD signal, x(t) represents the time-varying EEG signal and $e^{-j\frac{2\pi }{T}kt}$ is the negative exponential function. After performing DFT on the EEG signal, this paper uses the signal representation in the FD X(k) to calculate the PSD $f_{psd}$ and energy feature $f_{energy}$ of the α band and β band respectively, as the FD features of EEG:(5)$$f_{psd}=\frac{1}{T}{\left|X(k)\right|}^{2},f_{energy}={\left|X(k)\right|}^{2}$$

The final obtained subset of FD features: $F_{fd}=\left\{f_{psd}^{\alpha },f_{psd}^{\beta },f_{energy}^{\alpha },f_{energy}^{\beta }\right\}$.

#### 2.3.3. Time-Frequency Domain Features

Due to the susceptibility of EEG signals to various interferences, and considering that wavelet transform possesses strong noise resilience, it has become a highly effective method for extracting features from EEG signals [26]. This paper calculates the discrete wavelet energy as the Time-Frequency Domain (TFD) feature. The definition of Discrete Wavelet Transform (DWT) is:(6)$$W_{a,b}=\sum_{t=1}^{T}x(t)\psi_{a,b}(t)$$
where $W_{a,b}$ is the wavelet coefficient, $\psi_{a,b}$ is the wavelet basis function, a and b represent the frequency resolution and time translation, respectively. x(t) is decomposed into finite layers using the Mallat algorithm, resulting in:(7)$$x(t)=x_{L}^{A}(t)+\sum_{j=1}^{L}x_{j}^{D}(t)=A_{L}+\sum_{j=1}^{L}D_{j}$$
where *L* is the number of decomposition layers, $A_{L}$ is the low-pass approximation component, and $D_{j}$ is the detail component at different scales. This paper then utilizes the db4 wavelet basis for 4-layer wavelet decomposition, choosing *D3* to represent β wave and *D4* to represent α wave. The wavelet energy of the two bands is then calculated as the TFD features, with the wavelet energy feature $f_{dwt}$ calculated as follows:(8)$$f_{dwt}={\sum_{j=1}^{T}\left(D_{ij}\right)}^{2}$$
where i represents the wavelet decomposition at the i-th layer. The final obtained set of TFD features is the wavelet energy features $F_{tfd}=\{f_{dwt}^{\alpha },f_{dwt}^{\beta }\}$ of the α and β bands.

#### 2.3.4. Spatial Domain Features

The CSP algorithm, as a spatial domain (SD)feature, has been widely applied in the processing of EEG signals for BCIs based on MI [27]. The core idea of the CSP algorithm is to use matrix diagonalization to determine an optimal set of spatial projection filters. This approach maximizes the variance difference between the two classes of signals, thereby generating feature vectors with enhanced discriminative capabilities. The single-trial EEG signal can be represented as a matrix X of $N_{ch}\times T$. Here, $N_{ch}$ denotes the number of channels. First, compute the mixed spatial covariance matrix:(9)$$C_{C}=\bar{C_{1}}+\bar{C_{2}},\;C_{i}=\frac{X_{i}X_{i}^{T}}{trace(X_{i}X_{i}^{T})}(i=1,2)$$
where i represents the MI category, ${\bar{C}}_{1}$ and ${\bar{C}}_{2}$ are the mean covariance matrices for the first and second types of imagined movements, respectively, *trace*(•) denotes the trace of the matrix, and $X_{i}^{T}$ represents the transpose matrix. Then, $C_{C}$ performs eigen-decomposition on the mixed spatial covariance matrix:(10)$$C_{C}=U_{C}\lambda_{C}U_{C}^{T}$$
where $\lambda_{C}$ is the diagonal matrix composed of the eigenvalues of the mixed spatial covariance matrix $C_{C}$ arranged in descending order, and $U_{C}$ is the eigenvector matrix of $C_{C}$. The calculation formula for the whitening matrix is as follows:(11)$$P=\lambda_{C}^{-\frac{1}{2}}U_{C}^{T}$$

Applying the whitening matrix to $C_{i}$, we obtain:(12)$$S_{i}=PC_{i}P^{T},S_{i}=B\lambda_{i}B^{T}(i=1,2)$$
where $\lambda_{1}+\lambda_{2}=I$ can be used to compute the projection matrix W as $W=B^{T}P$. Where W is the matrix of $N_{ch}\times N_{ch}$. The single-trial EEG signal can be decomposed using the projection matrix into:(13)$$Z_{i}=W\times X_{i}(i=1,2)$$

Transforming the EEG signal according to the above equation, the CSP feature $f_{p}$ that can be ultimately used for training the classifier can be calculated from $Z_{p}$.(14)$$f_{p}=\log(\frac{\mathrm{var}(Z_{p})}{\sum_{i}^{2m}\mathrm{var}(Z_{i})})$$

Extract CSP features for each sub-band. Finally, the EEG data is segmented into six frequency bands, and CSP features are extracted from each band. The frequency range is from 8 to 32 Hz, with an interval of 4 Hz. Features p are extracted from each sub-band as the SD features $F_{sd}$.

### 2.4. Feature Selection and Feature Rotation Transformation

To obtain features that are more advantageous for classification, this study employs Recursive Feature Elimination (RFE) [28] to filter the feature sets from each domain, thereby reducing the dimensionality of the features and enhancing the performance of the base classifiers. RFE is a backward search algorithm that follows the wrapper pattern, and its effectiveness is contingent upon the classifier utilized during the iterative classification process. Considering the advantages of Random Forest (RF) [29], such as high accuracy and robustness, we have chosen RF as the iterative classifier for RFE. The primary steps of the RFE-RF algorithm, which utilizes the RF iterative classifier, are as follows: first, the RF model is trained using the candidate feature set. $F=\{f_{1},f_{2},f_{3},\cdots ,f_{n}\}$ Express the importance measure of each feature in the model as $\left\{I_{1},I_{2},I_{3},\cdots ,I_{n}\right\}$, where $I_{1}>\cdots >I_{i}>I_{j}$, and rank the feature set $\left\{f_{1},f_{i},\cdots ,f_{j}\right\}$ based on importance. Then, remove the feature $f_{j}$ corresponding to the least importance $I_{j}$, resulting in a new feature subset $\left\{f_{1},f_{2},\cdots f_{i}\right\}$. Repeat the training, calculation, and elimination process until the optimal feature set $F_{rfe}$ with $N_{a}$ features is obtained. After RFE-RF feature selection, the significant feature subsets obtained for each domain feature are $F_{rfe}^{td},F_{rfe}^{fd},F_{rfe}^{tfd},F_{rfe}^{sd}$.

Rotation Forest is a classification algorithm with a local rotation strategy [30]. By preserving all principal component information, local rotation helps achieve higher performance [31,32]. This paper proposes a method for further optimizing multi-domain feature sets through rotation transformation. Specifically, we perform local rotation to reconstruct the feature space, which reduces the correlation between features while simultaneously enriching their representations, thereby enhancing their discriminative ability. The key steps are outlined below:

(1) Randomly divide the original feature space K into $F_{j}(j=\{1,2,\ldots ,K\})$ non-overlapping local subspaces, each containing $M=N_{a}/K$ features.

(2) Perform a 75% sample resampling process on $F_{j}$ to construct a new subspace $V_{j}$.

(3) Apply PCA transformation on $V_{j}$ to obtain the principal component coefficients $O_{j}$, where $O_{j}$ is a matrix M×M. The principal component coefficients obtained from each local subspace are denoted as $O=\{O_{1},O_{2},\ldots ,O_{K}\}$. Construct a coefficient rotation matrix R using the principal component coefficients O. The arrangement of the rotation matrix is as follows:$$R=\left[\begin{array}{cccc} O_{1} & \left[0\right] & \cdots & \left[0\right]\\ \left[0\right] & O_{2} & \cdots & \left[0\right]\\ \vdots & \vdots & \ddots & \vdots \\ \left[0\right] & \left[0\right] & \cdots & O_{K} \end{array}\right]$$

The obtained rotation matrices R are generated via PCA, which determines the optimal linear transformations for each feature subset. This method maximizes the variance of the rotated features and increases feature diversity, thereby enhancing classifier performance.

Reorder the columns in R according to the original data feature set to obtain the matrix R″.

(4) By performing local rotation on the significant features, the rotated features $F_{rotated}$ is obtained as $F_{rotated}=F_{rfe}\cdot R\prime \prime$.

Based on the research on Extended Space Forest [33], this study integrates and connects significant features and rotated features to obtain a composite feature subset $F_{com}=\left[F_{rotated},F_{rfe}\right]$. Here, $\left[\cdot ,\cdot \right]$ denotes the connection operation. These can form a promotive relationship, enhancing the discriminability and diversity of the features. Ultimately, four sets of composite feature sets are obtained as $F_{com}^{td},F_{com}^{fd},F_{com}^{tfd},F_{com}^{sd}$. The detailed process of rotation fusion is shown in Figure 3.

### 2.5. Stacking Ensemble Model

In previous research on MI tasks, researchers often relied on single-classifier models. Additionally, the application of ensemble learning was typically limited to homogeneous ensemble methods or simple voting strategies. To address these issues and fully leverage multi-domain features, this paper proposes a stacking model framework that combines the predictions of classifiers trained in each domain, thereby enhancing the accuracy and robustness of recognition tasks related to MI. Stacking ensemble is a method of ensemble learning that merges the predictions of several base learners and incorporates them with a meta-learner to enhance the model’s predictive performance [34]. This paper chooses Random Forest (RF) as the base classifier in the first layer. The composite feature sets of various domains are used to train the base classifier RF, resulting in the respective base training model $B_{td},B_{fd},B_{tfd},B_{sd}$. Using time-domain features as an example, the output of the trained base classifier can be represented as a probability estimation vector $P^{td}={\left[p_{1}^{td},\ldots ,p_{C}^{td}\right]}^{T}$, where each element $p_{i}^{td}$ corresponds to the probability estimation for class i. Here, C denotes the number of target classes, and $P^{td}$ is a matrix of dimension N×C, where N is the number of training samples. In this way, four probability estimation vectors $P^{td},P^{fd},P^{tfd},P^{sd}$ can be obtained. In traditional stacking models used for MI tasks, the meta-classifier does not directly handle the original features, which may cause it to overlook some information in the original features, thus affecting the final performance of the meta-classifier. Therefore, this paper integrates original significant features from various domains and uses LDA for dimensionality reduction. The linear discriminant features $f_{LDA}=LDA(\left[F_{rfe}^{td},F_{rfe}^{fd},F_{rfe}^{tfd},F_{rfe}^{sd}\right])$ obtained from the dimensionality reduction are used as supplementary features for the meta-classifier. By maximizing the ratio between the inter-class variance and the intra-class variance of the original features, LDA can effectively extract the most discriminating linear combination for classification from the high-dimensional original features, avoiding the overfitting risk brought by the high-dimensional features, not only reducing the complexity of the model but also ensuring the retention of key information, thus improving the generalization ability of the model. The meta-classifier needs to consider composite optimization during the classification process. Therefore, this paper selects Multi-Layer Perceptrons (MLP) [35] as the meta-classifier model. We use the predicted probability estimates and linear discriminant features from each base classifier as inputs for the meta-classifier MLP. The meta-classifier is then trained to obtain the final classifier model h. The obtained probability estimation vector can be derived through the stacking model, denoted as P. Thus, $P=h(P^{td},P^{fd},P^{tfd},P^{sd},f_{LDA})$, ultimately choosing the category with the highest probability as the final prediction outcome.

## 3. Results and Discussion

### 3.1. Experimental Setup

Due to the non-stationary, low-amplitude, and low signal-to-noise ratio nature of EEG signals, as well as the instability of MI systems and their susceptibility to interference, EEG signals typically contain various forms of noise and interference, such as 50 Hz power line interference (with a noticeable interference pulse at 50 Hz in the EEG signal spectrum), as well as electrooculogram (EOG), electrocardiogram (ECG), and electromyogram (EMG) interferences. Considering the characteristics of MI tasks, where ERD and ERS phenomena occur during task execution, a fifth-order Butterworth zero-phase filter is applied to the EEG signals for bandpass filtering from 8 Hz to 30 Hz. For Datasets I and II, the data from 0.5 to 3.5 s after the visual cue in each experiment is extracted for subsequent feature extraction and classification. The number of features selected by RFE-RF is 56 and 35, respectively. For Dataset III, the data from 0.5 to 3 s after the visual cue in each experiment is extracted for subsequent feature extraction and classification, and only the MI task data for the left and right hand are selected for subsequent feature extraction and binary classification. We only performed a binary classification comparison between the left and right hand (L vs. R). The number of features selected by RFE-RF is 28. As for the selection of K, according to the discussion on the value of K in [32], that is, the selection of k has certain stability on the classification performance, so this paper chooses 7 as the value of K to facilitate the comparison and analysis of subsequent experiments. The parameters for the base classifiers and meta-classifiers used in this paper follow default parameter settings. To accurately assess the classification accuracy of the proposed method, a 5 × 5-fold cross-validation approach was employed in the experiments. The raw EEG dataset was divided into training and testing sets in a 4:1 ratio, and the experiments were repeated five times to minimize the adverse effects of randomness on the model evaluation.

### 3.2. Evaluation Metrics

To assess the model’s performance in a more comprehensive and accurate manner, we utilized several performance indicators, including Accuracy (*Acc)*, Precision (*Prec*), Recall (*Rec*), and F1 Score (*F*1). The calculation formulas for each indicator are as follows:(15)$$Acc=\frac{TP+TN}{TP+FP+TN+FN}$$(16)$$Prec=\frac{TP}{TP+FP}$$(17)$$Rec=\frac{TP}{TP+FN}$$(18)$$F1=2\times \frac{Prec\times Rec}{Prec+Rec}$$
where *TP*, *TN*, *FP*, and *FN* represent True Positives, True Negatives, False Positives, and False Negatives, respectively.

### 3.3. Classification Results

To illustrate the classification results of the method proposed in this study and to assess the effectiveness of using Random Forest as the base classifier, we compared our classification outcomes with those obtained using Support vector machine (SVM) [36] with RBF kernel (SVM-rbf) and Linear SVM (SVM-linear) as base classifiers. Table 1, Table 2 and Table 3 present the classification results of the proposed method on Datasets I, II, and III, utilizing RF, SVM-rbf, and SVM-linear as base classifiers, respectively. In order to provide a clearer visualization of the classification performance across different base classifiers, Figure 4 presents the mean confusion matrix for all datasets.

The results presented in Table 1, Table 2 and Table 3 indicate that both the RF and SVM perform well as base classifiers in this model. Among them, the RF classifier consistently achieved the highest average classification accuracy across Datasets I, II, and III. In Dataset I, the average classification accuracy of Random Forest was higher than that of the SVM-rbf and SVM-linear by 1.5% and 2.21%, respectively. Notably, RF achieved the highest classification accuracy and F1 score among all subjects except for subject aw. In Dataset II, RF outperformed the other two classifiers by 1.75% and 5.5% in average classification accuracy, demonstrating the best performance across all subjects. In Dataset III, RF improved the average classification accuracy compared to SVM-rbf and SVM-linear by 1.56% and 2.41%, respectively. Particularly for subject A01, RF’s classification accuracy was higher by 3.41% and 4.15%, indicating that RF, as a base classifier, also shows significant advantages in classification tasks for individual subjects. In summary, the RF classifier demonstrated high accuracy and stability as the foundational classifier in this study, exhibiting exceptional performance in classification tasks across the majority of subjects. At the same time, it can also be seen from Figure 4 that RF classifier achieved higher classification accuracies for both the left and right hands compared to SVM with RBF and linear kernels. Additionally, it maintained a lower error rate, demonstrating superior classification performance.

We also evaluated the running time of the proposed method through experiments. The training time for Dataset I was 20.46 s, with a test time of 5.12 s; for Dataset II, the training time was 9.2 s and the test time was 2.3 s; and for Dataset III, the training time was 12.27 s, with a test time of 3.1 s. Consequently, the time required for each trial across the three datasets was 0.092 s, 0.058 s, and 0.053 s, all of which are below 0.1 s. Therefore, this study has practical application significance.

In this study, we introduced the REF-RF method as a part of the proposed method feature selection process. In order to verify the influence of different numbers of features selected by RFE-RF on classification performance, Figure 5 shows the trend of classification accuracy with the number of features in the three datasets.

The classification results from the three datasets indicate that as the number of features increases, the classification accuracy tends to rise. However, after reaching a certain threshold, the accuracy stabilizes or even slightly declines. For instance, in Dataset II, when the number of features is relatively low, the classification accuracy is below 70%. As the number of features increases, the accuracy peaks at 89%. Yet, beyond this point, further increases in the number of features do not lead to any improvement in accuracy. At the same time, it can be seen from Figure 5 that the optimum number of features corresponding to each data set is different. For example, the optimum number of features in Dataset I ranges from 50–60, while the optimum number of features in Dataset III ranges from 25–30. This highlights the importance of feature selection in enhancing model performance while also suggesting that an excessive number of features may lead to overfitting.

### 3.4. Verification of the Effectiveness of Feature Rotation Transformation

To verify whether local rotation transformations can enhance feature separability, this study employed the same base classifier to train and classify three types of features: significant features (SF), rotated features (RotF), and composite features (CF) across various domains. Table 4, Table 5 and Table 6 present the classification accuracy results for different feature types in Datasets I, II, and III, respectively. To provide a clearer representation of the classification performance for each feature type, we calculated the sum and average of the classification results for each feature type across all domains for each subject. Figure 6a illustrates the average classification results for significant features, rotated features, and composite features from nine subjects in Datasets I and II, while Figure 6b displays the corresponding classification results for all nine subjects in Dataset III.

Based on the results presented in Table 4, Table 5 and Table 6, it is evident that the rotated features obtained through local rotation transformation significantly outperform the significant features across all domains, with a notable classification accuracy improvement of 6.25% in the FD of Dataset II. When comparing each subject individually, the rotated features achieve average classification accuracy increases of 3.21%, 3.1%, and 1.32% across the three datasets. These findings suggest that local rotation transformations effectively enhance feature discriminability by preserving all principal component information and reconstructing the feature space. Notably, when the significant feature set and the rotated feature set are combined to form a composite feature set, the classification accuracy does not decrease despite the increase in feature dimensions. On the contrary, for most subjects, the classification accuracy improves compared to that of the rotated features alone. In particular, for subjects A01 and A05, the classification accuracy significantly increased by 6.4% and 5.44%, respectively. Additionally, in Dataset II, all subjects except for subject g experienced classification accuracy improvements of over 5%. This enhancement may be attributed to the non-overlapping information carried by the locally rotated features and significant features, allowing the classifier to access a more diverse and useful set of information, thereby enhancing classification performance. Overall, applying rotation transformations to significant features can effectively improve the classification performance. From the classification results of significant features in different domains across three datasets, it is evident that for different subjects, the classification performance of features in each domain varies. For instance, temporal domain features perform best for subject aa, FD features perform best for subject al, and SD features perform best for subject g. Therefore, for MI classification, it is crucial to fully utilize information from different domain features for different subjects. To facilitate the visualization and comparative analysis of significant and rotated feature sets within the high-dimensional feature matrix, this study employs t-distributed Stochastic Neighbor Embedding (t-SNE) [37] to project the feature distributions into a two-dimensional space. Figure 7 displays the t-SNE plots of the feature distributions for the significant feature set, rotated feature set, and composite feature set for subject aa.

As shown in Figure 7, the rotated features exhibit significantly greater discriminability between the left and right categories compared to the significant features. Furthermore, the composite feature set, which combines these two types of features, contains richer EEG information, allowing for better categorization.

### 3.5. Verification of the Effectiveness of Multi-Domain Feature Stacking Strategy

To verify the effectiveness of the stacking method in leveraging the diverse information provided by multi-domain features, and to assess whether incorporating original features as supplementary inputs to the meta-classifier can yield effective classification results, this study conducted experiments comparing the fusion method, stacking method, and stacking fusion method against the optimal feature domains for each subject. The fusion method involved concatenating the significant features from each domain and inputting them into an MLP for training and classification. The stacking method utilized an RF classifier to separately train the significant features from each domain, subsequently feeding the probability estimation vectors from the RF into the meta-classifier MLP for training and classification. The stacking fusion method built upon the stacking approach by integrating the significant features from each domain, employing LDA to reduce dimensionality and obtain linear discriminant features, which were then input into the meta-classifier MLP for training. The optimal feature domain refers to the selection of the feature domain with the highest classification accuracy from multiple feature domains, based on the classification accuracies obtained through training. Figure 8a–c present the average classification accuracy results for Datasets I, II, and III using these three methods alongside the optimal feature.

As shown in Figure 8, the classification accuracy of the fusion method did not show a significant improvement compared to that of the optimal feature domains. Notably, the classification accuracy declined markedly for subjects av, f, and A06, with decreases of 12.5%, 7.5%, and 9.2%, respectively. Meanwhile, the feature fusion method also did not demonstrate significant advantages in robustness. This result suggests that simply concatenating features from different domains may not fully leverage the effective information from each domain; instead, it could reduce the model’s recognition capability due to the increase in redundant information. In contrast, the stacking method improved classification performance by 8.92%, 3.88%, and 5.15% across the three datasets, respectively. Furthermore, the average classification accuracy of the stacking method was consistently not lower than that of the optimal feature domains. Therefore, compared to simple feature fusion, the stacking method effectively integrated valuable information from various domains through ensemble learning, achieving effective complementarity of features. This approach reduced redundant information during the fusion process, thereby enhancing classification performance. This study also compared the stacking fusion method and the stacking method across the three datasets. The average classification accuracy of the stacking fusion method improved by 2.15%, 1.62%, and 1.88%, respectively, consistently outperforming the stacking method. By incorporating linear discriminant features as supplementary inputs into the meta-classifier, the model can effectively leverage the original information contained in the EEG features. Furthermore, LDA serves as a feature reduction technique, effectively preserving inter-class differences while reducing the dimensionality of the feature space. Therefore, by building upon the stacking method, the incorporation of linear discriminant features as supplementary inputs to the meta-classifier can significantly enhance the classification performance of the model.

To quantify the contribution of each domain feature in the stacking model classification, we trained the meta-classifier (MLP) using the prediction probability vectors from the base classifiers of each domain. We assessed each domain’s contribution by calculating the weights from the input layer to the hidden layer and from the hidden layer to the output layer in the MLP. Table 7 presents the specific contribution rates of each domain across different datasets.

As can be seen from Table 7, there is no significant difference in the contribution value of each domain to the classification performance, which indicates that each feature domain has a relatively consistent effect on improving the classification performance in the stacked integration model, and there is no obvious dominant feature domain. At the same time, it can be seen from Table 7 that spatial features show the highest classification contribution degree on all data sets, which also reflects that spatial features have certain advantages in EEG classification and can provide more differentiated data information.

### 3.6. Comparison with Other Methods

To verify the effectiveness of the proposed method in the MI classification task, we compare it with other MI classification methods. For Datasets I and II, we use the Ensemble Regularized Common Spatial Spectrum Patterns (Ensemble RCSSP) model [38], the Selective Extract of the Multi-View Time-Frequency Decomposed Spatial (S-MVTFS) method [39], and the Common Time-Frequency-Spatial Patterns (CTFSP) method [40]. The average classification accuracy of the test set is shown in Table 8. For Dataset III, we compare the proposed method with the Time Window-Tikhonov Regularization CSP-Filter Bank (TW-TRCSP-FB) method [41], the S-MVTFS model, the compact convolutional neural network for EEG (EEGNet) model [42] and Sliding Window with Longest Consecutive Repetition based CSP method (SW-LCR) [43]. The average classification accuracy of the test set is shown in Table 9.

From Table 8 and Table 9, it can be observed that the proposed method exhibits the best performance across the three datasets. For Dataset I, compared to the three mentioned methods, the proposed method improves the average classification accuracy by 7.92%, 6.01%, and 2.78%, respectively. Similarly, for Dataset II, it enhances the average classification accuracy by 14.13%, 3.47%, and 2%, respectively, when compared to the same methods. Moreover, for Dataset II, our method shows higher classification accuracy for all subjects compared to other methods, indicating its excellent adaptability to individual differences. Among the aforementioned methods that use only single-domain features, the classification performance is relatively poor. These methods usually fail to fully capture the complexity and diversity of EEG signals, and due to individual differences among subjects, the classification accuracy for some subjects is very low. For example, the CTFSP method shows classification accuracy that is more than 20% lower than the proposed method for subjects av and b. This highlights the importance of effectively utilizing multi-domain features in MI tasks. For Dataset III, the proposed method improves the average classification accuracy by 10.98%, 6.24%, 6.78%, and 1.42%, respectively, compared to the other methods. Compared with the CTFSP, E-RCSSP, EEGNet, and SW-LCR methods, the approach proposed in this paper achieves a significant improvement in classification accuracy (*p*-value < 0.05). Notably, EEGNet is a classic deep learning algorithm for MI tasks. The proposed method not only surpasses the method in classification accuracy but also demonstrates a smaller standard deviation, indicating enhanced stability and robustness across subjects. Although the proposed method does not exhibit highly significant advantages over the other two approaches, it still achieves improvements in classification accuracy across Datasets I, II, and III. At the same time, from the perspective of each subject, subject al in Dataset I achieved over 95% classification accuracy across most methods, while subject “av” exhibited a marked decline in accuracy. Subjects b in Dataset II and A02 and A04 in Dataset III also exhibited a similar pattern. These findings highlight the individual differences inherent in MI tasks. Additionally, the average classification accuracy of different data sets is different, indicating that different experimental environments and experimental paradigms have an impact on the quality of EEG signals. These results underscore the effectiveness, stability, and robustness of the proposed method, demonstrating its ability to effectively combine multi-domain features and stacking ensemble, thereby enabling efficient recognition of MI tasks.

## 4. Conclusions

In this study, we propose a novel framework based on multi-domain feature rotation transformation and stacking ensemble for classifying MI tasks. To fully leverage the effective information embedded in multi-domain features and achieve information complementarity across domains, we employed a stacking ensemble approach to integrate the prediction results from different domains. Additionally, to enhance the separability of the extracted EEG features, we introduced rotation transformations for feature space reconstruction. The proposed method was experimentally evaluated on three datasets and compared with state-of-the-art MI techniques. The experimental results demonstrate that our approach exhibits superior classification performance in MI tasks. In future work, we can increase the diversity of the ensemble framework by incorporating additional domain features as well as classifier combinations. Furthermore, we can select the most suitable feature and classifier combinations for different subjects to achieve optimal classification performance.

## Figures and Tables

**Figure 1 brainsci-15-00050-f001:**
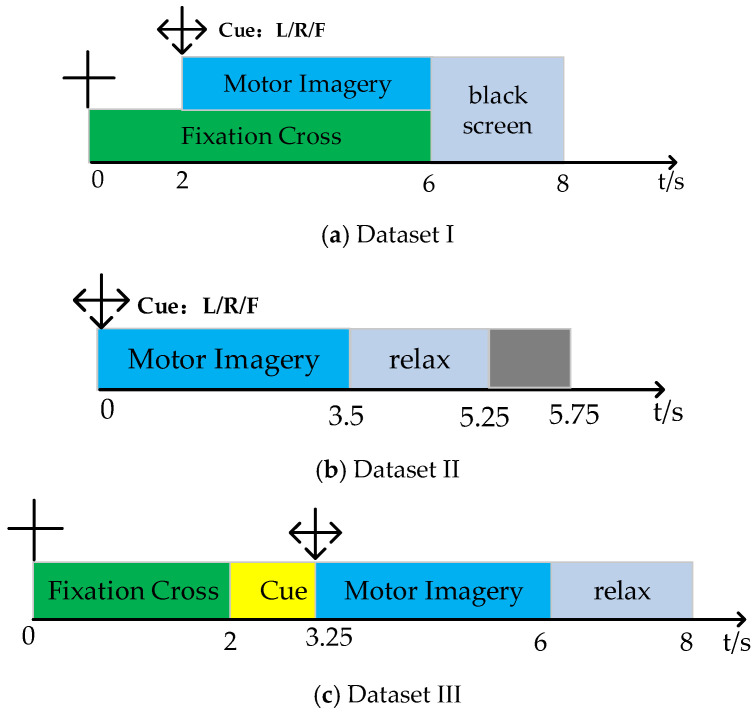
Dataset timeline.

**Figure 2 brainsci-15-00050-f002:**
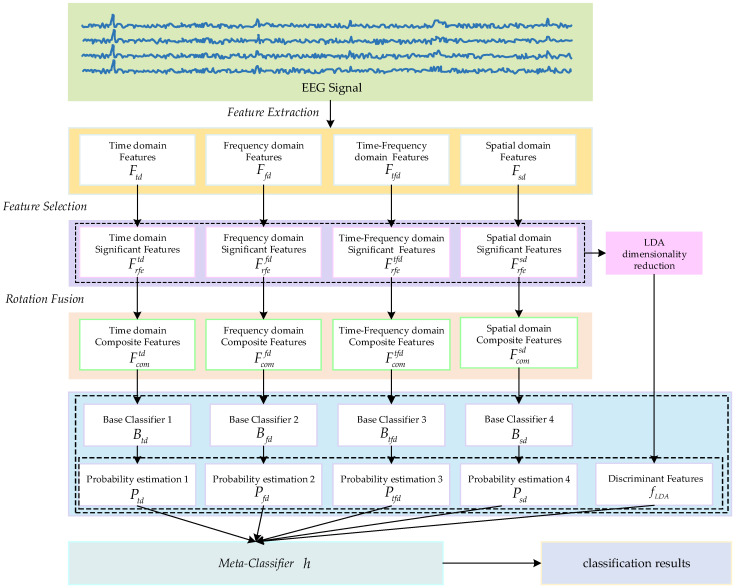
The method diagram proposed in this article.

**Figure 3 brainsci-15-00050-f003:**
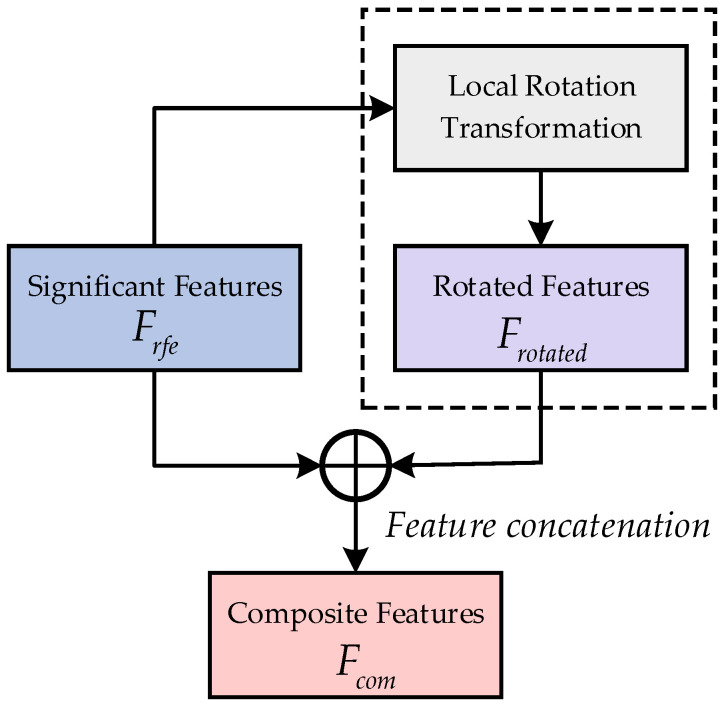
Rotation fusion process diagram.

**Figure 4 brainsci-15-00050-f004:**
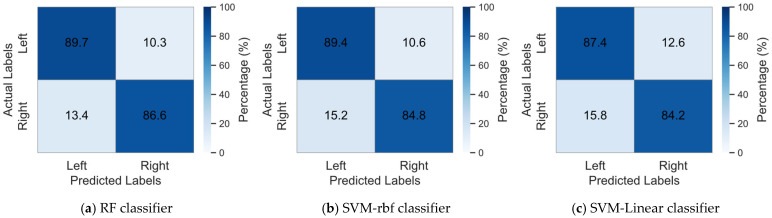
Mean confusion matrix of different base classifiers on three datasets.

**Figure 5 brainsci-15-00050-f005:**
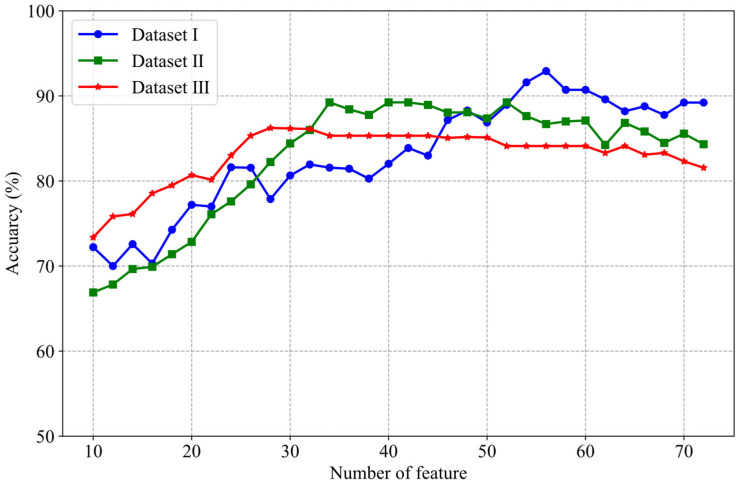
Features selection performance of the RFE-RF method for different numbers.

**Figure 6 brainsci-15-00050-f006:**
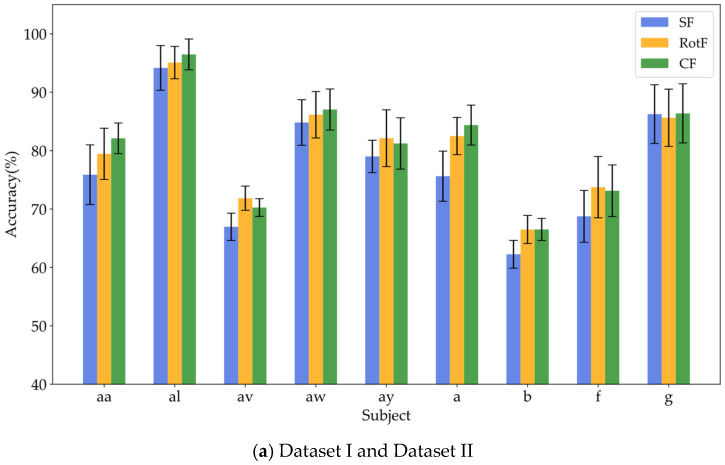
Visualization of classification performance comparison among different feature types.

**Figure 7 brainsci-15-00050-f007:**
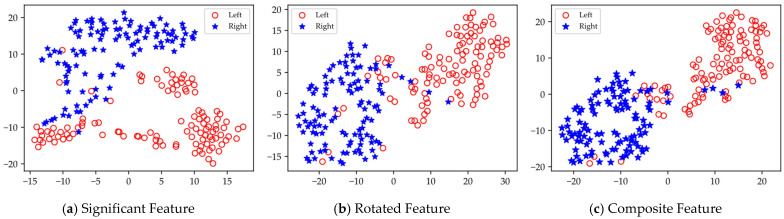
Visualization of feature distribution comparison of subject aa in dataset I.

**Figure 8 brainsci-15-00050-f008:**
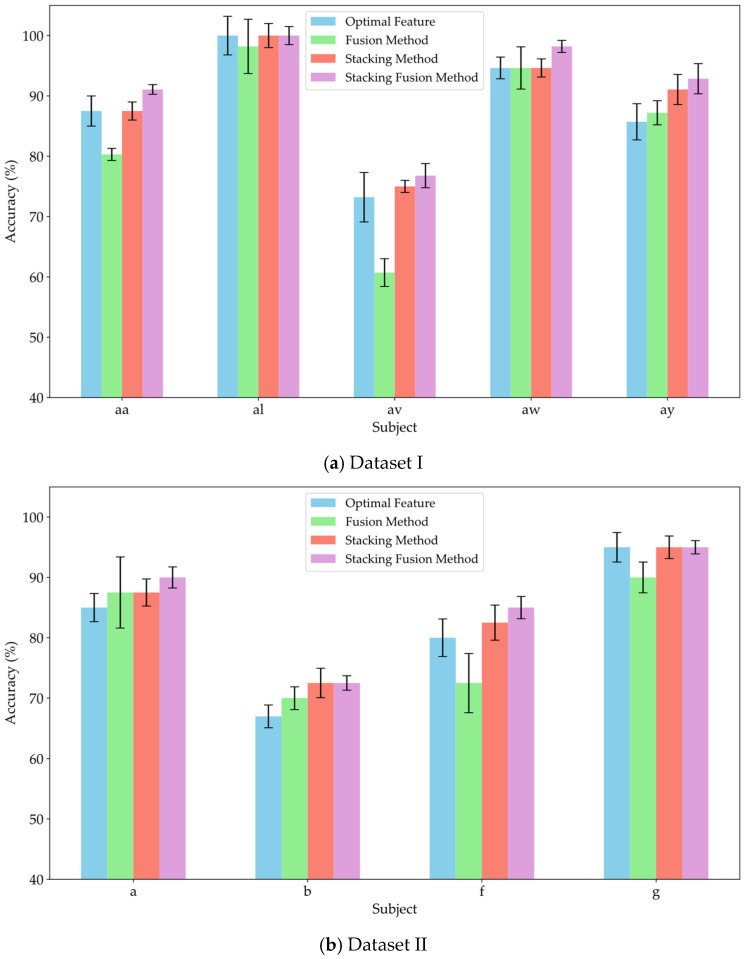
Comparison of classification accuracy between feature fusion and stacking strategies.

**Table 1 brainsci-15-00050-t001:** Classification results for dataset I using RF, SVM-rbf, and SVM-linear. Bold font is used to indicate the best outcome.

Subject	Classifier											
	RF				SVM-rbf				SVM-Linear			
	*Acc*	*Prec*	*Rec*	*F*1	*Acc*	*Prec*	*Rec*	*F*1	*Acc*	*Prec*	*Rec*	*F*1
aa	**92.85**	**91.16**	**96.05**	**93.61**	89.28	86.67	92.85	89.76	87.5	90.09	88.23	89.16
av	**80.35**	**75.75**	**83.33**	**79.54**	76.78	75	78.57	76.79	78.57	82.14	76.67	79.41
al	**100**	**100**	**100**	**100**	100	100	100	100	98.21	96.42	100	98.24
aw	96.42	**96.29**	**96.29**	**96.29**	**98.21**	96.42	100	98.21	94.64	96.15	92.59	94.37
ay	**95**	**93.98**	95.36	94.67	94.64	92.85	**96.29**	94.57	94.07	93.41	**96.29**	**94.85**
mean	**92.92**	**91.28**	**94.17**	**92.73**	91.42	90.28	93.01	91.87	90.71	91.53	90.75	91.21

**Table 2 brainsci-15-00050-t002:** Classification results for dataset II using RF, SVM-rbf, and SVM-linear. Bold font is used to indicate the best outcome.

Subject	Classifier											
	RF				SVM-rbf				SVM			
	*Acc*	*Prec*	*Rec*	*F*1	*Acc*	*Prec*	*Rec*	*F*1	*Acc*	*Prec*	*Rec*	*F*1
a	**91.50**	84.36	96.64	**90.5**	90.00	**85**	94.4	89.7	87.5	84.2	88.89	86.54
b	**77.50**	75.33	79.00	**77.17**	**77.50**	72.72	**80**	76.36	75	75	75	75
f	**90.50**	**90.50**	88.57	**89.53**	87.5	86.36	90.47	88.42	82.5	77.27	**89.47**	83.37
g	**97.50**	93.83	95.78	**94.81**	95	**94.73**	**94.73**	94.73	90	85.7	**94.73**	90.21
mean	**89.25**	**86.01**	**90.00**	**88.00**	87.50	84.70	89.90	87.30	83.75	80.54	87.02	83.78

**Table 3 brainsci-15-00050-t003:** Classification results for dataset III using RF, SVM-rbf, and SVM-linear (L vs. R). Bold font is used to indicate the best outcome.

Subject	Classifier											
	RF				SVM-rbf				SVM-linear			
	*Acc*	*Prec*	*Rec*	*F*1	*Acc*	*Prec*	*Rec*	*F*1	*Acc*	*Prec*	*Rec*	*F*1
A01	**91.38**	**87.97**	**95.21**	**91.59**	87.94	83.74	85.84	88.74	87.23	82.5	94.72	88.61
A02	**72.41**	**72.21**	81.97	**77.09**	69.71	64.28	**88.73**	74.55	69.01	63.63	78.73	71.18
A03	94.6	92.58	94.64	93.61	**96.47**	**98.52**	94.36	**96.4**	94.36	94.01	94.54	94.20
A04	**76.72**	72.82	80.86	76.84	74.64	70.11	**85.91**	**77.21**	73.23	**81.13**	60.56	69.35
A05	83.13	85.09	84.5	**84.79**	**85.11**	85.4	**88.18**	86.79	83.68	**86.36**	80.28	83.32
A06	**79.31**	**77.14**	84.6	80.87	78.87	73.56	90.14	**81.01**	75.35	69.14	**91.51**	78.78
A07	**87.93**	83.88	**91.36**	**87.62**	83.68	86.36	80.28	83.21	82.39	**87.09**	76.05	81.2
A08	**97.01**	96.67	**96.89**	**96.78**	95.07	94.32	96.02	95.17	96.47	**97.52**	95.36	96.4
A09	**93.85**	**94.97**	92.11	**93.54**	90.84	88.18	93.38	90.78	92.95	91.86	**95.59**	93.33
mean	**86.26**	**84.81**	89.12	**86.97**	84.70	82.72	**89.20**	85.98	83.85	83.69	85.26	84.04

**Table 4 brainsci-15-00050-t004:** Comparison of classification accuracy among different feature types in Dataset I. Bold font is used to indicate the best outcome.

Subject	Time Domain			Frequency Domain			Time-Frequency Domain			Spatial Domain		
	SF	RotF	CF	SF	RotF	CF	SF	RotF	CF	SF	RotF	CF
aa	87.50	**91.07**	89.28	66.07	73.21	**78.50**	69.64	**73.21**	78.57	80.30	80.30	**82.10**
av	73.21	**78.57**	76.78	58.92	**69.60**	67.85	62.50	**66.07**	65.00	**73.21**	**73.21**	71.40
al	92.85	92.85	**96.42**	100.00	100.00	100.00	94.64	**96.46**	**96.64**	89.20	91.00	**92.85**
aw	76.78	80.30	**82.1**	85.71	**87.50**	**87.50**	**82.14**	80.35	**82.14**	94.64	**96.46**	**96.46**
ay	**73.21**	69.60	69.60	80.35	**89.28**	87.50	76.78	**80.35**	**80.35**	85.71	**89.28**	87.50
Mean	80.71	82.48	**82.83**	78.21	83.92	**84.27**	78.93	**82.86**	81.97	84.61	86.05	**86.6**

**Table 5 brainsci-15-00050-t005:** Comparison of classification accuracy among different feature types in Dataset II. Bold font is used to indicate the best outcome.

Subject	Time Domain			Frequency Domain			Time-Frequency Domain			Spatial Domain		
	SF	RotF	CF	SF	RotF	CF	SF	RotF	CF	SF	RotF	CF
a	80.00	80.00	**85.00**	70.00	**87.50**	**87.50**	67.50	**75.00**	**75.00**	85.00	87.50	**90.00**
b	57.00	62.50	**65.00**	65.00	**70.00**	67.50	60.00	**62.50**	**62.50**	67.00	71.00	**71.00**
f	60.00	**65.00**	**65.00**	65.00	67.50	**70.00**	70.00	**75.00**	72.50	80.00	**87.50**	85.00
g	**72.50**	**72.50**	**72.50**	**87.50**	**87.50**	**87.50**	90.00	87.50	**90.50**	**95.00**	**95.00**	**95.00**
Mean	67.38	70.00	**71.88**	71.88	**78.13**	**78.13**	75.00	75.00	**75.13**	81.75	**85.25**	**85.25**

**Table 6 brainsci-15-00050-t006:** Comparison of classification accuracy among different feature types in Dataset III (L vs. R). Bold font is used to indicate the best outcome.

Subject	Time Domain			Frequency Domain			Time-Frequency Domain			Spatial Domain		
	SF	RotF	CF	SF	RotF	CF	SF	RotF	CF	SF	RotF	CF
A01	60.34	62.50	**67.00**	74.14	**81.13**	**81.13**	72.41	74.52	**75.21**	84.48	**86.12**	85.05
A02	50.07	**53.11**	55.22	56.53	**57.95**	56.89	61.37	54.22	53.52	**56.53**	55.47	**56.53**
A03	79.85	79.85	**80.21**	76.92	79.12	**80.95**	83.15	84.45	**86.24**	**95.83**	**95.83**	**95.83**
A04	53.07	**57.45**	54.82	57.45	56.89	**59.64**	56.03	59.64	59.64	68.40	69.45	**70.00**
A05	72.10	72.10	**78.26**	73.45	74.63	**77.89**	74.63	75.55	**78.51**	69.92	71.42	**74.63**
A06	55.55	**58.13**	**58.13**	54.88	58.13	58.13	63.72	65.92	**68.14**	72.92	75.00	**76.10**
A07	74.43	**76.81**	**76.81**	76.38	79.06	**79.42**	61.99	**62.96**	61.99	81.91	**83.60**	**83.60**
A08	81.91	83.02	**84.13**	73.89	76.89	**78.22**	72.56	72.56	**78.51**	86.21	**87.86**	87.50
A09	75.37	73.48	**76.51**	75.37	74.62	**77.45**	85.98	**87.54**	87.18	90.28	90.50	**92.36**
Mean	66.97	68.49	**70.12**	68.78	70.94	**72.19**	70.20	70.82	**72.10**	78.50	79.47	**80.18**

**Table 7 brainsci-15-00050-t007:** Classification contribution of each feature domain on different datasets (%).

	Time Domain	Frequency Domain	Time-Frequency Domain	Spatial Domain
Dataset I	24.64	25.06	24.43	25.87
Dataset II	23.58	25.18	24.85	26.39
Dataset III	23.90	24.78	24.79	26.53

**Table 8 brainsci-15-00050-t008:** Comparison of average classification accuracy with different methods on Dataset I and II. Bold font is used to indicate the best outcome.

Subject	Methods			
	CTFSP [40]	E-RCSSP [38]	S-MVTFS [39]	Proposed Method
aa	86.07	82.14	85.36	**92.85**
av	52.14	68.87	79.64	**80.35**
al	98.57	96.42	95.71	**100.00**
aw	96.07	**98.21**	95.00	96.42
ay	92.14	88.88	**95.00**	**95.00**
Mean ± Std	**85.00** ± **16.96**	**86.91** ± **11.94**	**90.14** ± **6.5**	**92.92** ± **6.70**
a	86.50	87.50	89.00	**91.00**
b	53.50	74.50	72.00	**77.5**
f	66.50	91.00	**92.00**	90.5
g	93.50	90.50	95.00	**97.5**
Mean ± Std	75.00 ± 15.88	85.66 ± 6.7	87.13 ± 8.91	**89.13** ± **7.25**
*p*-value	0.013	0.015	0.110	**-**

**Table 9 brainsci-15-00050-t009:** Comparison of average classification accuracy with different methods on Dataset III (L vs. R). Bold font is used to indicate the best outcome.

Subject	Methods				
	EEGNet [42]	SW-LCR [43]	TW-TRCSP-FB [41]	S-MVTFS [39]	Proposed Method
A01	63.30	86.81	91.67	**92.50**	91.38
A02	61.20	64.58	56.25	72.14	**72.41**
A03	87.30	96.53	95.83	**96.43**	94.60
A04	63.50	69.45	68.65	**80.36**	76.72
A05	**87.00**	60.42	56.25	69.29	83.13
A06	56.00	72.92	75.00	73.93	**79.31**
A07	86.20	79.87	81.25	87.50	**87.93**
A08	82.50	95.84	97.22	**97.86**	97.01
A09	90.50	93.06	93.06	93.57	**93.85**
Mean ± Std	75.28 ± 13.86	80.02 ± 13.45	79.48 ± 15.39	84.84 ± 11.11	**86.26** ± **8.25**
*p*-value	0.012	0.028	0.061	0.442	**-**

## Data Availability

The three datasets used in this study, namely BCI Competition III Data Sets Iva, BCI Competition IV Dataset Sets I, and BCI Competition IV Dataset 2a, are all publicly available. The dataset links are as follows: https://www.bbci.de/competition/iii/, accessed on 15 June 2024 and https://www.bbci.de/competition/iv/, accessed on 15 June 2024.

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
