# Peer review of "Motor Imagery EEG Classification Based on Multi-Domain Feature Rotation and Stacking Ensemble"

_brainsci, 2025, doi:10.3390/brainsci15010050_

Round 1

Reviewer 1 Report

Comments and Suggestions for Authors

The presented work is devoted to the analysis of EEG data for the classification of human motor activity. The authors use three different datasets to test their approach. The novelty of the solution lies in the selection of the main EEG features, their layout and the use of an ensemble of classifiers. The basic classifiers (random forest) are combined within the Stacking ensemble. The authors provided a detailed description of datasets and data processing procedures, their approach, and the classification quality assessment metrics used. In the experimental part, they compared their approach with several alternative solutions, moreover, this comparison was multi-stage, involving comparisons of basic classifiers, ensembles, and evaluation of the influence of features.  Thus, the experimental part is presented quite fully and thoroughly with the presentation of the necessary comparisons. The number of references to related research is sufficient. As a result, I positively evaluate this work.

The only comment that does not have a significant impact may be the usefulness of demonstrating the error matrix for the classification problem being solved (not necessarily for all cases, perhaps only within the framework of the experiment in Section 3.3), since such a graphical representation can further simplify the understanding and evaluation of the results.

Reviewer 2 Report

Comments and Suggestions for Authors

This paper presented a very well-crafted method for improving the classification of EEG data into different types of motor imagery. The research in this line bears significance to the BCI field. This work certainly has this merit given the detailed report of the sound and sophisticate methods they developed. I commend the efforts of the authors, and I recommend clarifying the following points listed below.

1)      Rotation appears to be one of the essential steps in the algorithm. However, it is not entirely clear to the reader what kind of rotation is the best, or how to achieve the best rotation, or, was the optimal rotation automatically found by the method? I hope the authors could improve the explanation in this part.

2)      The second point concerns some inconsistencies and unclarity in the result. As shown in Figure 5, the accuracy for classifying the Composite Feature would probably achieved a value close to 100%, as there was only one blue circle that was mixed with the red one. However, this high accuracy was not reflected in the table. Could the authors check on this? Also, what do the two axes mean?

Reviewer 3 Report

Comments and Suggestions for Authors

Thanks for the invitation to review this work. The article presents an approach for motor imagery EEG classification by employing a multi-domain feature rotation transformation combined with a stacking ensemble framework. However, potential issues including insufficient explanation of methodological details, lack of generalizability analyses, unclear contributions of individual classifiers, limited statistical validation, and missing real-time feasibility considerations could be further improved.

1.      The authors introduce local rotation transformation but lack details on how non-overlapping subspace selection impacts results or the variability caused by this process.

2.      The role of LDA in mitigating overfitting in the meta-classifier is underexplained, particularly regarding the reuse of original features.

3.      The contributions of individual base classifiers in the stacking ensemble are not quantified, leaving the impact of each domain on overall accuracy unclear.

4.      The benefits of composite feature fusion in handling imbalanced classes or noise remain insufficiently explained.

5.      The results lack analysis of subject variability (noise levels or signal quality) and its effect on performance across datasets.

6.      The rationale for selecting the number of features after RFE and its impact on performance is unclear.

7.      Differences in accuracy compared to other methods could be influenced by preprocessing or parameter tuning, but these aspects are not explicitly discussed.

8.      Statistical significance testing for metrics like accuracy, precision, recall, and F1 score is absent, making the reported improvements less robust.

9.      Latency, computational cost, and real-time feasibility are not considered, which are critical for practical applications.

10.   In figure 1, timelines are clear but do not explain how fixation, cue, motor imagery relate to feature extraction. In figure 3, the "fusion operation" lacks a mathematical or algorithmic explanation. In figure 4, bar plots lack confidence intervals or error bars, and legends could use more precise labels (e.g., "SF" as "Significant Features"). The t-SNE Visualization (Figure 5) shows improved discriminability but do not justify the choice of t-SNE over PCA or LDA, nor confirm result consistency across runs. The determination of "optimal feature domains" in figure 6 is not explained.

11.   Consider including citations of Exploration, 2024, 4, 20230146.

Round 2

Reviewer 3 Report

Comments and Suggestions for Authors

Thanks for the invitation to review this work. The authors have tried to solve the previous concerns, and the article is recommended for publication after careful proof check:

1. In introduction, “multi-domain feature rotation transformation” and “stacking ensemble” can be explained concisely for clarity.

2. In method, "Time Domain (TD) features are the relevant information or statistical data contained in EEG signals along the time dimension." is a bit awkward. Consider rephrasing: "Time Domain (TD) features are statistical measures derived from EEG signals that reflect relevant information along the time dimension."

3. In result, "The experimental results demonstrate that the method proposed in this paper outperforms existing methods in terms of classification accuracy and robustness." Please detail the “existing methods” for clarity.

4. Ensure that abbreviations, such as MI (Motor Imagery), are consistently defined when first mentioned and are used consistently throughout the manuscript.
